# Rapid Decrease in Dextrose Concentration After Intra-Articular Knee Injection: Implications for Mechanism of Action of Dextrose Prolotherapy

**DOI:** 10.3390/biomedicines13020350

**Published:** 2025-02-04

**Authors:** Kenneth Dean Reeves, Jordan R. Atkins, Clare R. Solso, Chin-I Cheng, Ian M. Thornell, King Hei Stanley Lam, Yung-Tsan Wu, Thomas Motyka, David Rabago

**Affiliations:** 1Independent Researcher, Roeland Park, KS 66205, USA; 2Medical School, DeBusk College of Osteopathic Medicine, Lincoln Memorial University, Harrogate, TN 37752, USA; jordan.atkins@imunet.edu; 3Independent Researcher, Mission, KS 66202, USA; csolsogood@gmail.com; 4Department of Statistics, Actuarial and Data Science, Central Michigan University, Mt. Pleasant, MI 48859, USA; cheng3c@cmich.edu; 5Department of Internal Medicine, Pappajohn Biomedical Institute, Roy J. and Lucille A. Carver College of Medicine, University of Iowa, Iowa City, IA 52242, USA; ian-thornell@uiowa.edu; 6The Hong Kong Institute of Musculoskeletal Medicine, Hong Kong 999077, China; 7Department of Family Medicine, The Chinese University of Hong Kong, Hong Kong 999077, China; 8Department of Family Medicine, The University of Hong Kong, Hong Kong 999077, China; 9Center for Regional Anesthesia and Pain Medicine, Chung Shan Medical University Hospital, Taichung 402, Taiwan; 10Center for Regional Anesthesia and Pain Medicine, Wan Fang Hospital, Taipei Medical University, Taipei 110, Taiwan; 11Department of Physical Medicine and Rehabilitation, Tri-Service General Hospital, School of Medicine, National Defense Medical Center, Taipei 11490, Taiwan; crwu98@gmail.com; 12Integrated Pain Management Center, Tri-Service General Hospital, School of Medicine, National Defense Medical Center, Taipei 11490, Taiwan; 13Department of Research and Development, School of Medicine, National Defense Medical Center, Taipei 11490, Taiwan; 14Department of Osteopathic Manipulative Medicine, School of Osteopathic Medicine, Campbell University, Lillington, NC 27506, USA; motyka@campbell.edu; 15Department of Family and Community Medicine, Penn State College of Medicine, Hershey, PA 17003, USA; drabago@pennstatehealth.psu.edu

**Keywords:** glucose, prolotherapy, monosaccharide transport proteins

## Abstract

Background: D-glucose (dextrose) is used as a 5000–25,000 mg% solution in the injection-based pain therapy known as dextrose prolotherapy (DPT). The number of peer-reviewed clinical trials supporting its use is growing. However, the mechanism of action is unknown, limiting further research. A commonly expressed theory is that hyperosmotic dextrose injection induces inflammation, initiating a healing-specific inflammatory cascade. In vitro study models have used continuous exposure to high concentration dextrose. But the rate of dextrose clearance after intra-articular injection, and, therefore, the duration of exposure of tissues to any particular dextrose concentration, remains unknown. We therefore determined the rate of dextrose concentration diminution in one human participant’s knees after intra-articular dextrose knee injection. Method: In this pre–post N-of-1 study, the first author (KDR), a well 70-year-old male without knee-related pathology, injected his own knees with 30 mL of 12,500 mg% dextrose on three occasions; performed serial aspirations of 1.2 mL of intra-articular fluid from 7 to 360 min post-injection; and assessed synovial dextrose concentration. Dextrose clearance kinetics were determined using Minitab and GraphPad Prism software. Results: Dextrose concentration dropped rapidly in all three trials, approximating an exponential or steep S curve. A third order chemical reaction pattern was found, suggesting factors other than dilution or glucose transporter activity, such as rapid diffusion of dextrose across the synovial membrane, may have contributed to the rapid drop in dextrose concentration. Conclusion: This pre-post N-of-1 study shows that, after intraarticular injection of 30 mL of 12,500 mg% dextrose injection into a well knee, the concentration of dextrose diminished rapidly, suggesting that intra-articular cells, tissue, and anatomic structures are exposed to an initially high dextrose concentration for a very short time. This likely affects the mechanism of action of DPT and should inform in vitro study methods.

## 1. Introduction

The growing prevalence of non-cancer chronic pain has prompted calls for development of new therapies [1]. Several injection techniques delivering therapeutic solutions to specific pain sites have emerged, one of which is called prolotherapy [2]. Developed in the early twentieth century, its treatment protocols were refined in the 1950s [3]. The name of the technique stems from the observation that, in early animal model studies, tissues receiving injections responded by growing larger, or “proliferating” [3]. Clinical and research practice has focused on the effects of injections with the right (dexter; D)-rotated isomer of glucose (D-glucose, or dextrose) that is metabolizable in humans. The technique is often referred to as dextrose prolotherapy (DPT), and involves the injection of dextrose into entheses and adjacent joints to treat musculoskeletal pain and dysfunction [4]. Favorable meta-analyses assessing DPT using a hyperosmolar dextrose injectate have been published for eight common neuromusculoskeletal conditions [5,6,7,8,9,10,11,12].

A 1991 paper about potential mechanisms suggested osmotic shock at the injection site as a potential mechanism of action [13]. High osmolar dextrose was thought to cause removal of water from living cells, subsequent local tissue trauma, and induction of the inflammatory cascade with a net anabolic tissue effect [4,13]. The osmotic shock theory had a profound influence on subsequent in vitro study design and interpretation of results. Researchers have commonly treated dextrose as a steady-state solute by exposing a variety of human and animal cells to fixed concentrations of dextrose in culture over 24 h [14,15]. Dextrose at a fixed level for 24 h initiates cell death or apoptosis with as little as a 1000 mg% (mg per deciliter) dextrose concentration [16]. Researchers assessing dextrose prolotherapy for a wide variety of conditions have included this theory of dextrose mechanism in their clinical publications, characterizing DPT as “injection of an irritant” that initiates an inflammatory response, activating the inflammatory cascade, with resultant tissue proliferation that favors healing [17].

However, evidence is accumulating in four areas that challenge the osmotic shock mechanism theory of DPT. The first is histologic. In contrast with in vitro reports of findings from 24 h cell cultures, in vivo post-injection histologic studies do not support an inflammatory mechanism of DPT. Researchers report the absence of neutrophil invasion 1, 2, 4, 8, and 12 weeks after 10,000 mg% dextrose injection in rabbits without needle trauma [18], and few leukocytes 24 h after 15,000 mg% dextrose injection, similar to needling with or without injection in rats [19]. The second is use of lower concentration (5000 mg%) isosmotic dextrose, which is known to be non-inflammatory [20], in two therapeutic dextrose injection techniques related to DPT: perineural injection treatment (PIT) [21,22] and ultrasound-guided hydrodissection of entrapped nerves [23,24,25]. Positive clinical trials of both techniques suggest that inflammation is not, or not the only, working mechanism for dextrose in DPT. The third is solution chemistry, including the rapid clearance of solutes such as dextrose across the cells lining the synovium of rabbits [26]. The fourth is dilutional effects; symptomatic knees often have an increased synovial fluid volume, which would serve to rapidly dilute the injected dextrose to non-inflammatory concentration levels.

A clear understanding of tissue-level dextrose concentration in the minutes and hours post-injection would inform understanding of the possible mechanistic options for DPT and, hence, optimize in vitro study designs. However, little is known about the fate of intra-articular dextrose in conventionally performed DPT; in particular, the dextrose concentration over time in joint spaces subsequent to injection. We therefore sought to measure dextrose concentration over time after intra-articular injection in a human model. Our hypothesis was that intraarticular dextrose concentrations will drop rapidly after injection, and that cellular exposure to high initial concentration dextrose will be brief.

## 2. Materials and Methods

This study took place from 27 November to 18 December 2023. This was a pre–post, N-of-1 study, during which multiple post-injection synovial fluid dextrose concentration curves were produced after injection into non-osteoarthritic knees. The lead author (KDR) was the single participant and performed intra-articular injections on his own knees, followed by serial aspirations at pre-determined time intervals. The aspirated fluid was assessed for dextrose concentration, and concentration–time curves were developed.

### 2.1. Injection Procedure and Injectate

In the participant’s clinic exam room, and with the participant in a semi-reclined position with knee in extension with a pillow under the knee, a quadriceps contraction was performed. An ultrasound linear array probe (GE NextGen LOGIC eR8, Boston, MA, USA) was positioned transversely across the distal thigh just above the patella, at the level of the suprapatellar pouch, and demonstrated the absence of fluid in the suprapatellar pouch (Figure 1A). After application of 4% chlorhexidine for antisepsis, intra-articular self-injection of 30 mL of 12,500 mg% dextrose (3 × 10 mL syringes) was performed via an inferolateral approach with knee flexed at 90 degrees. The needle was then removed from the intra-articular space and a quadriceps contraction under ultrasound was repeated with knee once again in extension, demonstrating a fluid-filled suprapatellar pouch, and confirming appropriate placement of the injectate (Figure 1B). A flexible catheter (Terumo Medical Corporation SR*FF2051, Somerset, NJ, USA) with a Luer lock was then inserted in the suprapatellar pouch to obtain serial synovial fluid samples (Figure 1C).

### 2.2. Aspiration Procedure

Per laboratory protocol, each synovial fluid aspirate volume was 1.2 mL; at 7 min in Trial 1 (after a delay for catheter placement) and at 10 min in Trails 2 and 3. Aspirations were then performed serially at 30 min intervals from 240–360 min, for up to 13 samples in Trial 1, and at less frequent intervals during Trials 2 and 3 when a catheter was not in place. Trials 1 and 2 were performed on the right knee and Trial 3 was on the left knee. Standard intraarticular injection aseptic precautions were used; due to the small risk of knee infection, similar to that of knee arthroscopy [27], two grams of oral cephalothin were administered to the participant 30 min prior to the dextrose injection in each of three trials.

### 2.3. Determination of Dextrose Concentration

Our primary outcome measure was dextrose concentration in mg%. Timed 1.2 mL aspirates were drawn into syringes, transported to a Clinical Laboratory Improvement Amendments (CLIA)-certified laboratory (Labcorp, Overland Park, KS, USA), and were kept in a refrigerated state at 3–8 degrees centigrade until analysis. For those dextrose values above the measurement range limit of standard laboratory equipment (<750 mg%; Cobas c 701 module that performs photometric assay tests for a wide range of analyses, Roche Diagnostics, 124 Grenzacherstrassse, Basel, Switzerland), 1 to 10 part dilutions (synovial fluid to normal saline) were performed using a standardized protocol [28] until the dextrose values fell within the readable range. Only dextrose within the interstitial fluid is measured by the Glucose Hexokinase Gen.3 (GLUC3) test (Cobas); intracellular dextrose is not assessed. The GLUC3 test catalyzes glucose to glucose-6-phosphate by hexokinase and oxidases glucose-6-phosphate in the presence of NADP to gluconate-6-phosphate. The rate of NADPH formation during the reaction, which is directly proportional to the glucose concentration, is measured photometrically [29,30].

### 2.4. Analysis of Dextrose Concentration Curves

In order to better characterize the pattern by which dextrose concentration decreased within each trial, a best fit curve statistical analysis was performed using Minitab version 21.4 (Minitab, LLC, State College, PA, USA, www.minitabl.com, accessed 29 January 2025) and the Pearl-Reed logistic curve [31]). This method of analysis takes a variety of geometric curve types and analyzes statistically to determine the curve with the best (lowest) mean absolute percent error (MAPE), which is then chosen as the best fit curve [32].

Data from all three trials were then combined and fit with the integrated form of the best fit rate equation using GraphPad Prism software version 10 (GraphPad Software, 225 Franklin Street, Fl. 26, Boston, MA, USA) for the purpose of characterizing how the rate of reaction depends on the concentration of reactants. The data were best represented by a third-order reaction:$$=\frac{{\left[glucose\right]}_{t_{0}}}{\sqrt{1+2*k*t*{\left[glucose\right]}_{t_{0}}^{2}}}$$
where *t* is elapsed time, ${\left[glucose\right]}_{t_{0}}$ is the initial glucose concentration, ${\left[glucose\right]}_{t}$ is the glucose concentration measured at time point *t*, and *k* is the rate constant obtained by the fit.

## 3. Results

### 3.1. Demographics

The participant was a 70-year-old male Physical Medicine and Rehabilitation clinician scientist, 1.8 m tall, with a weight of 76 kg (BMI 23.5 m/kg^2^), and no history of knee trauma. His Kellgren–Lawrence scores were 0 for each knee, with a plain film series for each knee read by a board-certified radiologist as “No acute fracture or malalignment. The joint spaces are normal. No joint effusion. Superior patellar enthesophytes”.

### 3.2. Injection Trial Time Points

Trial 1 was performed on 27 November 2023 on the participant’s right knee (Table 1). The first aspiration was performed 7 min following initiation of intraarticular injection, and every half hour through 270 min (Table 1). The intracath failed before all aspirations were completed, requiring its removal, and direct aspiration from the suprapatellar pouch was required to obtain the last four samples. An initial examination of dextrose levels across time in Trial 1 showed that the dextrose level approached, but did not reach, the normal synovial fluid baseline of 100 mg% by 270 min. Trials 2 and Trial 3 were conducted on 18 December 2023; both left and right knees were injected. Given the failure of the intracath in Trial 1, we elected to forego intracath placement in Trials 2 and 3. In order to avoid a large number of direct aspirations while still collecting sufficient data for curve mapping, each knee was aspirated at two early time points (10 and 30 min after dextrose injection), and two later time points (180 and 360 min after dextrose injection).

### 3.3. Dextrose Concentration Results from Trials 1–3

Dextrose concentration values, and the corresponding contribution of dextrose alone to total osmolality, are presented in Table 1. All values were measured by standard laboratory techniques, except for pre-injection (time 0) and 2-min values. We estimated a time 0 (pre-injection) dextrose concentration of 100 mg%, which is understood to be the approximate fasting dextrose level in both human serum and intraarticular synovial fluid [34]. Two minutes is the approximate time to complete an injection. Due to the time required for catheter placement and taping, a sample was not obtained at that time, and dextrose concentration was estimated by a simple proportional dilution calculation (Table 1 footnotes). In Trial 1, the laboratory-assessed dextrose concentration fell rapidly, from 4016 mg% at 7 min, to 416 mg% at 270 min. In Trials 2 and 3, a normal synovial fluid baseline dextrose concentration was reached by 360 min, suggesting that all the excess dextrose beyond baseline levels had exited the synovial fluid by that time and was no longer measurable.

Figure 2 shows the results from modeling the rate of decrease in dextrose concentration across the three trials of intraarticular injection of 30 mL of 12,500 mg% dextrose.

For the first trial, the S curve model approximated the observed data most closely, as confirmed by the best (lowest) mean absolute percent error (MAPE). This was likely facilitated by having the most points available and including an earlier measurement point. The large mean absolute deviation (MAD) and mean square deviation (MSD) were due to the discrepancy between the initial concentration and the predicted initial concentration. There was a marked drop for the first 7 min. The rapid decrease in concentration continued until 60 min, after which the drop slowed. In the second and third trials, the growth (exponential) model approximated the observed data the most closely, with a marked drop to the first measurement point at 10 min, a continued rapid drop until 30 min, and a slower drop after 30 min. The fact that the dextrose concentration dropped in either an exponential or steep S curve pattern strongly suggests rapid movement of dextrose out of the synovial fluid.

Figure 3: Combined data from all three trials, fit to a rate law.

The data can be approximated by a third-order reaction. These kinetics suggest that the local rise in dextrose concentration in the synovial fluid with injection cannot be cleared by local cells that consume dextrose with first-order kinetics [35]. This interpretation is consistent with the finding that tissues clear glucose via GLUT 1 and 4 with a Michaelis–Menten constant (Km; the substrate concentration at which an enzyme operates at half of its maximum velocity) of 4.0 mM [36]. Cells at the site of injection will be unable to consume substantially more glucose, despite the steep gradient into the cell, as they rapidly reach the saturation point for GLUT transport.

## 4. Discussion

The primary findings of this N-of-1 study are that dextrose concentration declined rapidly after knee intraarticular injection of 30 mL of 12,500 mg% dextrose (in the range used for DPT), in an exponential or steep S curve pattern consistent with third-order kinetics. The initial dilution effect, plus a limited GLUT transporter function due to a near-saturation state [36], cannot explain third-order kinetics. Concentration-dependent dextrose movement across the synovial membrane into the synovial vessels may occur [26,37]. These findings suggest that the often-referenced theory that dextrose induces ongoing osmotic shock and subsequent inflammation is not, or is not the only, mechanisms of action for DPT.

The contribution of dextrose to total osmolality became minimal by 30 min across all three trials, varying from 48 to 128 mM. A transient (for several seconds), hyperosmotic effect may cause cells to dump their vesicular contents [38], including inflammatory and cell-damaging cytokines, such as tumor necrosis factor alpha (TNFα), by synovial cells and chondrocytes [39]. However, given the absence of histologic evidence for inflammation effects other than from needling in vivo [18,19], and the ability of dextrose to block and reverse the inflammatory effects of TNFα and likely other inflammatory cytokines [40,41], a brief release of inflammatory cytokines is likely of limited importance; an inflammatory state resulting from a brief hyperosmolar state is unlikely to be a significant effect of DPT injection or a major mechanism for its action.

The theory of osmotic-shock-induced inflammation has impacted human, animal, and in vitro study designs and outcomes to date. Clinical study designs in humans have been least impacted by the osmotic shock theory, other than resulting in a common requirement that trial participants abstain from taking NSAIDs during clinical trials. Participant adherence to DPT protocols during those trials does not appear to have been adversely affected; the efficacy of DPT compared with that of different control interventions has been reported in multiple meta-analyses [5,6,7,8,9,10,11,12,42,43]. However, in clinical practice, the rigid avoidance of NSAID use may be overzealous and result in unnecessary discomfort during flare periods.

The focus of in vivo animal studies has been moderately impacted by the osmotic shock theory in that a major research focus has been to explore expected post-injection inflammation histologically. However, data have been mixed; in a rat model, the injection of 15,000 mg% dextrose into rat medial collateral ligaments did not result in more inflammatory cells (leukocytes and macrophages) than needling without dextrose [19]. In a rabbit model, inflammatory cells were found to be completely absent at all time points after injection of 10,000 mg% dextrose with minimal needling trauma into subsynovial (tendon sheath) tissue [18].

In vitro study designs have been most impacted by the osmotic shock theory. Standard approaches have involved culturing cells for 24 h in various concentrations of dextrose. Cell death has been found to increase proportionately to dextrose concentration over a 24-h culture period. Human synovial fibroblasts exposed to 594 mg% dextrose produced unfavorable/catabolic cytokines [44], human tenocyte viability decreased slightly after exposure to 1600 mg% dextrose [14], mouse chondrogenic (ATDC5) cells became apoptotic (decreased or ceased their metabolic activity) at or above dextrose concentrations of 6300 mg% [45], mouse fibroblast cell viability decreased markedly (by 25% to 100%) with exposure to 5000 mg% to 10,000 mg% dextrose, respectively [16], and human neuroblastoma (Sh-SY5Y) cell viability decreased by 78% to 100% with exposure to 5000 mg% to 15,000 mg% dextrose [46]. The result of using high, constant, dextrose levels is that (1) the design does not mimic clinical conditions, and (2) the subsequent results, including cell death, may lead to misleading recommendations for further basic or clinical science, such as “relatively low concentrations of dextrose should be used in DPT to avoid excessive inflammation” [16,46]. In short, some in vitro experimental designs may have little to do with the effects of dextrose injection in vivo.

### 4.1. Clinical Applications and Research Implications

A few researchers have used much lower dextrose concentrations for in vitro research, while maintaining the approach of culturing cells up to 24 h. Favorable anabolic polypeptides were produced by ATDC5 mouse chondrogenic cells in 325 mg% dextrose [47,48]. Human neuroblastoma (Sh-SY5Y) cells, rendered apoptotic with production of unfavorable cytokines by exposure to tumor necrosis factor alpha (TNF-α), experienced restoration of normal metabolism and a marked reduction in unfavorable cytokines when exposed to 450 mg% dextrose [41], and this was replicated with similar findings, including a reduction in reactive oxygen species production [40]. Another potential strategy to simulate in vivo results by in vitro research would be to expose cells to high dextrose concentrations for no more than 10–30 min, followed by prompt replacement of culture medium with a culture medium known to be optimum for the cell in question. We also suggest that the term “hypertonic dextrose injection” be avoided because changes in volume of the surrounding synovial cells remain unknown without evidence of lysis [49,50]. The rapid clearance of dextrose may be analogous to the effect of intravenous administration of hyperosmolar dextrose, during which rapid removal of a supraphysiological dextrose concentration results in hypotonic plasma [51].

### 4.2. Strengths and Limitations

Limitations include the study design, that is, only one individual was studied. Generalizability of these findings is likely, however, as dextrose concentration curves were similar across both knees and three trials, initial dextrose dilution is proportional to synovial fluid volume, kinetics of the GLUT transporter are known and consistent across species [52], and diffusion coefficients across synovial membranes are consistent within species and increasingly characterized [37]. The fact that the first author and study participant are the same person is unusual, and introduces potential bias, but is not unique. All relevant IRB and consent protocols consistent with human subject research were followed, and objective lab analysis limits potential for bias. This aspect of the study is conceptually similar to that of Barry Marshall, who self-inoculated with *H. pylori* while exploring an infectious cause of gastritis [53], and later won a Nobel prize for this work [54]. A second limitation was our failure to obtain exact osmolality readings. We also did not obtain as many readings for dextrose levels in Trials 2 and 3 due to a non-functioning intracath device; use of a larger bore intracath is recommended in future studies to obtain more uniform data points.

In this study we injected a substantially higher volume of dextrose than is customary in the prolotherapy clinical community for intraarticular injection of the knee, and these knees were asymptomatic. The question therefore arises whether and how these results translate into clinical medicine. It is reasonable to expect that our findings were conservative in several respects, that is, concentration changes in arthritic or otherwise less healthy knees may be even faster than our results suggest. First, the mg load of dextrose injected into the knee in this study (3750 mg) was three times the milligram load typically injected intraarticularly in DPT of 1250 mg (5 mL of 25,000 mg% or 10 mL of 12,500 mg% dextrose), resulting in a long period in which the dextrose levels were meaningfully elevated compared to that of DPT. Second, Heilman et al. found that asymptomatic osteoarthritic knees had mean synovial fluid volumes of 24.2 ± 16.3 mL [33]. Injection of 5 mL of 25,000 mg% dextrose or 10 mL of 12,500 mg% dextrose, after mixing with the larger synovial fluid volume, would result in dextrose concentrations of 4000 and 3700 mg% immediately post-dilution, making the potential for a brief inflammatory response to hyperosmotic dextrose injection even more unlikely. Injection in the presence of an injured knee with inflammation would likely be into a larger synovial volume as well, due to the acute response to injury, and synovial penetration rates may vary unpredictably

### 4.3. Non-Osmotic Theories for Mechanisms of Action of DPT

If osmolar elevations of the injected dextrose solutions do not meaningfully contribute to our understanding of the mechanism of action of DPT, to what can we attribute the therapeutic efficacy of DPT reported across multiple neuromusculoskeletal conditions? Benefits of dextrose in neurogenic pain have been the primary focus of mechanism-related research. Correction of a perineural glycopenic state has been proposed [21], given the profound effect of glycopenia on increasing the firing rate of C fibers [55] and the speed of analgesia after perineural dextrose injection (seconds) [56,57]. A Substance P (SP)-elevating effect of dextrose injections has been suggested, given evidence of an analgesic effect of SP outside the spinal cord (in the brain or periphery) [58,59], evidence that the presence of SP was required for antinociceptive effects of dextrose injection in a murine model [60], and demonstration of SP elevation in the synovium 1 week after knee injection with dextrose [61]. A favorable effect of dextrose on neurogenic inflammation is supported by a potent protective effect of dextrose on nerve cells adversely affected by tumor necrosis factor-alpha (TNF-α) [40,41]. The mechanism of benefits for dysfunctional connective tissue, and, specifically, characteristics of obvious anabolic effects of dextrose such as proliferation of soft tissue [62,63] and chondrogenesis/chondroprotection in grade IV osteoarthritis [64,65], have not been well studied. Limitations include a lack of proprietary interest in dextrose, in vitro study designs that emphasized the inflammatory model, and the great complexity and interactions of cytokines. The importance of further in vitro studies incorporating our findings are requisite, and would inform further, more clinically relevant translational research.

## 5. Conclusions

The main finding of this N-of-1 pre-post study is that dextrose concentration, and the related dextrose contribution to hyperosmolarity dropped rapidly upon injection following a third order curve, suggesting at least three contributing factors. Movement of dextrose rapidly across the synovial membrane is likely of more importance than either dilution or transport into cells by GLUT transporters. Our findings suggest that, to optimally explore potential mechanisms of DPT, in vitro studies should reduce dextrose concentration levels and duration of exposure.

## Figures and Tables

**Figure 1 biomedicines-13-00350-f001:**
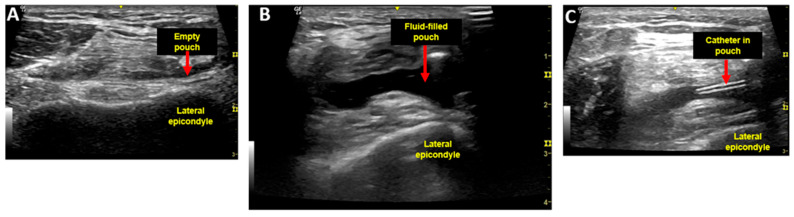
Ultrasound image of axial view just proximal to the patella to view the suprapatellar pouch with quadriceps contraction to force excess synovial fluid into the pouch. (**A**) shows the pouch prior to intra-articular injection and without fluid; (**B**) shows the pouch immediately after intra-articular injection with fluid. (**C**) shows the intraarticular catheter within the pouch.

**Figure 2 biomedicines-13-00350-f002:**
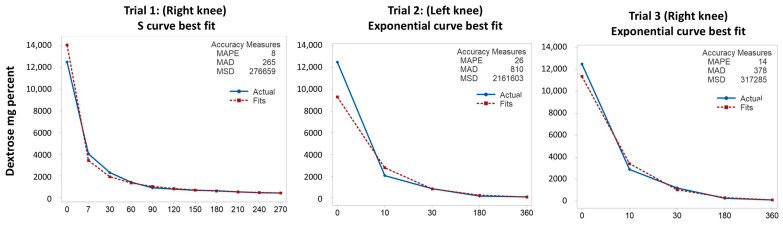
Best curve approximation for Trials 1–3 of intra-articular dextrose injection.

**Figure 3 biomedicines-13-00350-f003:**
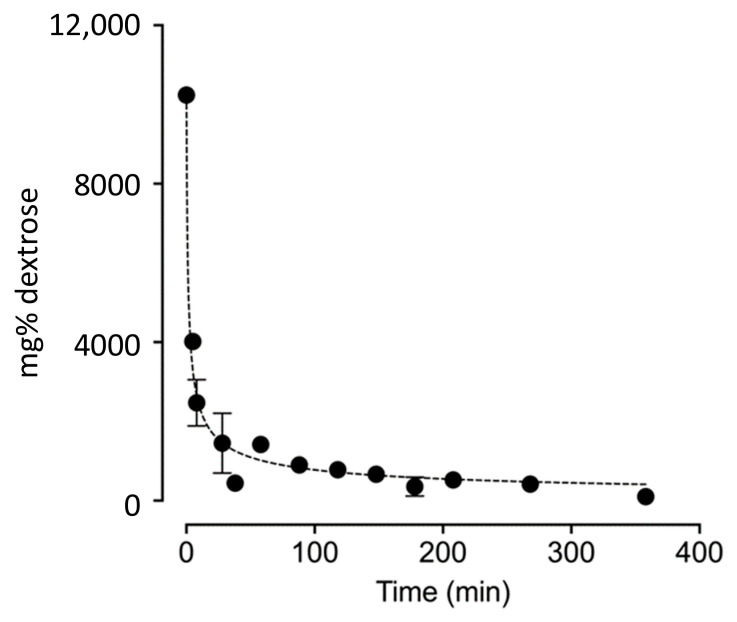
Combined data from all three trials, fit to a rate law.

**Table 1 biomedicines-13-00350-t001:** Dextrose levels in synovial fluid and corresponding dextrose osmolality (mM) at timed intervals after beginning injection of 30 mL of 12,500 mg% dextrose.

Minutes After Beginning Injection	Trial 1 R Knee mg% (mM)	Trial 2 R Knee mg% (mM)	Trial 3 L Knee mg% (mM)
0 Baseline synovial fluid dextrose (Estimated)	100 (6)	100 (6)	100 (6)
0 Injectate dextrose concentration	12,500 (695)	12,500 (695)	12,500 (695)
2 Estimated dilution effect ^1^	10,237 (641)	10,237 (641)	10,237 (641)
7	4016 (223)		
10		2883 (160)	2057 (114)
30	2303 (128)	1195 (66)	856 (48)
60	1417 (79)		
90	899 (50)		
120	778 (43)		
150	667 (37)		
180	630 (35)	258 (14)	178 (9)
210	523 (29)		
240	442 (25)		
270	416(23)		
300			
330			
360		102 (6)	102 (6)

^1^ The estimated 2-min dextrose concentration was calculated as follows: the mean synovial fluid volume found by Heilman et al. in asymptomatic non-osteoarthritic knees [33] was 6.7 ± 2.3 mL. An estimated time 0 volume of 6.7 mL was considered reasonable in the absence of any visible fluid in the suprapatellar pound with quadriceps contraction. To estimate the dextrose concentration at the time of injection completion, after an initial dilution effect, we used the following calculations: 12,500 mg% (12,500 mg per 100 mL) × 30 mL = 3750 mg (total mg of dextrose injected into knee); 100 mg% (100 mg per 100 mL) × 6.7 mL = 6.7 mg (total mg of dextrose contributed by 6.7 mL of synovial fluid); 3750 mg + 6.7 mg = 3757 mg (total mg of dextrose combining injectate and synovial fluid dextrose); 3757 mg/36.7 mL (total mg of dextrose divided by the total volume of injected dextrose plus synovial fluid) = 102.37 mg/mL = 10,237 mg/100 mL = 10,237 mg% (mg% of dextrose estimated at 2 min).

## Data Availability

Original glucose concentration data reports from the laboratory are available by request from the corresponding author.

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
