# Peer review of "Rapid Decrease in Dextrose Concentration After Intra-Articular Knee Injection: Implications for Mechanism of Action of Dextrose Prolotherapy"

_biomedicines, 2025, doi:10.3390/biomedicines13020350_

Round 1

Reviewer 1 Report

Comments and Suggestions for Authors

This is an unusual study to find a consenting subject for knee catheterization.  The study has limited data from one subject and may be suitable for an abstract or a very brief report.  Results for an injured knee with inflammation may differ somewhat, probably with even more rapid diffusion of glucose out of the synovial space.

Presentation of X-ray image does not add much.

Multiple presentation-- a table and multiple figures of essentially one set of glucose data.  Could present as one table or figure.

Unsure whether the photograph of specimens shows much other than  slight increase of blood.

Results are essentially as would be expected.

Very extensive review of background and discussion for such limited data

Discussion about osmotic effects but no measurement of osmolality or electrolyte concentrations that would have actually showed the effects of the glucose infusion.

SI units of measure would be mg/L for glucose.

The manuscript might be considered in substantially shortened form.

It is curious why there should be so many authors for such a small study.  Maybe a cover letter describing what the authors actually did and where the study was performred.

The one small redeeming factor is that this is a difficult study to perform from ethical consideration and obtaining a consenting subject.

Author Response

Comment 1: This is an unusual study to find a consenting subject for knee catheterization. The study has limited data from one subject and may be suitable for an abstract or a very brief report. Results for an injured knee with inflammation may differ somewhat, probably with even more rapid diffusion of glucose out of the synovial space.

 Response 1: We agree it is quite unusual for a physician to use themselves as the subject of a clinical study. But it is not unique; we note that Barry Marshall PhD infected himself with H pylori in order to explore an infectious cause for GERD; his work led directly to larger studies, a clear causal relationship, and later a Nobel prize in medicine. Clinical and basic science research on dextrose prolotherapy (DPT) are core professional interests of several of the authors. We were indeed concerned about finding a consenting participant and IRB approval for same. After discussion with ICMS-IRB, and the understanding that informed consent is an essential element of IRB approval, consent was granted for the lead author to serve as a test participant as a completely informed participant.

In lines 332-334 we clarified as , “All relevant IRB and consent protocols consistent with human subjects research were followed, and objective lab analysis limits potential for bias. This aspect of the study is conceptually similar to that of Barry Marshall who self-inoculated with H. pylori while exploring an infectious cause of gastritis[53], and later won a Nobel prize for this work[54].

Regarding the study question itself, we are also concerned that the basic science of prolotherapy rests on untested assumptions; one of those is the subject of the current project. We believe that the results of this N-of-1 study merit a stand-alone paper despite the small number of data points, and that these methods and results will inform larger studies.

We appreciate the importance of addressing potential differences in the presence of an injured knee with inflammation. Comparison of injured to non-injured knees is a fascinating consideration and should be part of future work.

We altered lines 341-354 in this resubmission as follows: (changes underlined)  “It is reasonable to expect that our findings were conservative in several respects, that is, concentration in arthritic or otherwise less healthy knees may be even faster than our results suggest. First, the mg load of dextrose injected in the knee in this study (3,750 mg) was three times the milligram load typically injected intraarticularly in DPT of 1,250 mg (5 mL of 25,000mg% or 10 mL of 12,500mg% dextrose), resulting in a long period during which dextrose levels will be meaningfully elevated compared to that of DPT. Second, Heilman et al. found that asymptomatic osteoarthritic knees had mean synovial fluid volumes of 24.2 +/- 16.3 ml [34]. Injection of 5 mL of 25,000mg% dextrose or 10 mL of 12,500mg% dextrose, after mixing with the larger synovial fluid volume, would result in dextrose concentrations of 4,000 and 3,700mg% immediately post-dilution, making the potential for a brief inflammatory response to hyperosmotic dextrose injection more unlikely. Injection in the presence of an injured knee with inflammation would likely be into a larger synovial volume due to the acute response to injury, and synovial penetration rates may vary unpredictably.”

Thank you for pointing out that we need to clarify why these results are likely to be generalizable. While dextrose concentration curves were very similar across both knees and three trials, we understand that the role of small study, no matter how consistent the data, is mainly to stimulate further research.

We altered lines 324-329 as follows: “Limitations include the study design, that is, only one individual was studied.  Generalizability of these findings is likely, however, as dextrose concentration curves  were similar across both knees and three trials, initial dextrose dilution is proportional to synovial fluid volume, kinetics of GLUT transporter are known and consistent across species, and diffusion coefficients across synovial membranes are consistent within species and increasingly characterized[52].”

Comment 2: Presentation of X-ray image does not add much.

Response 2:  We agree. The X-Ray image (Figure 2 was removed).    

Comment 3: Multiple presentation-- a table and multiple figures of essentially one set of glucose data. Could present as one table or figure.

Response 3: We mostly agree and have deleted Figure 3. However, Figure 4 includes MAPE values which relate to how closely each trial curve followed either S or exponential curve patterns and these curve patterns characterize  the initial speed of decline in dextrose concentration. In addition, figure 5 combined the data from all three trials, and fit that data to a rate law, which is critical to show that the kinetics of the decrease in dextrose concentration could not be explained by diffusion or GLUT transport alone. So, we have deleted figure 3, but  did not delete figures 4 and 5.

Comment 4: Unsure whether the photograph of specimens shows much other than slight increase of blood.

Response 4: If this does not add much, our readers may feel the same, and we removed figure 6.

Comment 5: Results are essentially as would be expected.

Response 5: We hypothesized that concentration decrease would be swift; but this has not been demonstrated. These results are novel. What was unexpected to us, and to those that have performed and researched dextrose prolotherapy, was the sheer speed of reduction in dextrose concentration, and the understanding that GLUT transport kinetics would not likely allow for that, suggesting rapid diffusion through the synovial membrane into synovial capillaries as a likely way to account for a 3rd order reaction. These results will also be novel to the prolotherapy basic science field, as prior work has often been predicated on a continuous concentration.

Comment 6: Very extensive review of background and discussion for such limited data

Response 6: We agree that the review is extensive in the context of a relatively small (though remarkably consistent) dataset, but consider it necessary for two reasons. First, prolotherapy remains an alternative therapy; detailed review is requisite because some readers will be unfamiliar with it. Second, as noted in the Discussion, these internally consistent results contradict the current mechanism theory of dextrose prolotherapy, which is that cells and tissue are exposed to high concentration dextrose for a lengthy time period. In vitro models have followed suit. If corroborated with larger studies, these results will inform basic science designs, interpretation of clinical trials, and future clinical trials.  

Comment 7: Discussion about osmotic effects but no measurement of osmolality or electrolyte concentrations that would have actually showed the effects of the glucose infusion.

Response 7: We agree with the limitation of not measuring exact osmolarity levels over time in parallel with dextrose values.  We have listed no osmolality values except in Table 1, with those based on the osmolality of the dextrose component only. We clarified lines 196 and 197 as follows, “Dextrose concentration values, and the corresponding contribution of dextrose to total osmolality, are presented in Table 1”.

Since we don’t have total osmolality values, we eliminated all speculation about the potential for a hypoosmolar state to develop. The only statement we made in this revision about osmolarity changes was  in the discussion in lines 263-264 as, “The contribution of dextrose alone to total osmolality became minimal by 30 minutes across all three trials, varying from 48-128 mM.” 

Comment 8: SI units of measure would be mg/L for glucose.

Response 8:  Thank you for encouraging us to stick with a single measure for dextrose concentration throughout the manuscript for consistency. Although clinicians speak in terms of percentage of dextrose in clinical practice, we have presented this revision using mg% which is mg/dL by standard. We are assuming there was a typo above and the reviewer meant to say mg/dL.  

Comment 9: The manuscript might be considered in substantially shortened form.

Response 9: We agree with some additional brevity and have cut  3 figures from the manuscript as described in responses 2-4 above.  However, we believe the literature review, explanation of the osmolar shock/inflammatory theory, the kinetics observed, a Discussion that addresses effects of the current model inflammatory mechanism theory of DPT on basic and clinical science research and clinical practice, and a section that addresses the question of “How else might DPT work?” are important in the context of these results and will be of interest to readers in this growing research and clinical area.

Comment 10: It is curious why there should be so many authors for such a small study. Maybe a cover letter describing what the authors actually did and where the study was performed.

Response 10:; The inclusion of all listed authors  is consistent with the four primary criteria of the International Committee Medical Journal Editors (ICMJE), namely substantial contributions to the conception or design of the work; or the acquisition, analysis, or interpretation of data for the work; AND drafting the work or reviewing it critically for important intellectual content; AND final approval of the version to be published; AND agreement to be accountable for all aspects of the work in ensuring that questions related to the accuracy or integrity of any part of the work are appropriately investigated and resolved.

The authors comprised several teams that made critical input. We described their involvement areas in the standard way with the submission, but it may be helpful to describe their involvement as teams.

On-site: (Kansas City) Clare Solso, R.N., assisted with practical clinical issues such as material procurement, choice of catheter placement, securing catheter, timing of samples, use of anesthetic for numbing, positioning for aspiration, sample management and laboratory site agreements. Jordan Atkins, med student, assisted with IRB submission, IRB presentation, injection, procuring of references for the study, and location of appropriate laboratory.

  1. Dean Reeves, M.D., was the subject and P.I.

Basic Science and Clinical Research Advisor Team:

Yung-Twan Wu, M.D., and King Hei Stanley Lam, M.D., were both engaged in and published on an approach to basic science research that involved much lower concentrations of dextrose. Their different approach led to divergent findings of the neuroprotective effect of dextrose on human neural cells. Their teaching is being altered by their findings and inspired this study. They gave valuable input on design and from a manuscript standpoint, and critical perspective on the importance of these findings for human basic science and clinical research.

Thomas Motyka, D.O. is a colleague with specific research interest in osteoarthritis and DPT research; he reviewed the manuscript, provided editorial input and, with his access to animal research, perspective on how these findings would impact animal basic science research.    

Statistics and Physiology: Chin-I Cheng, Ph.D., analyzed data and provided analysis of the nature of dextrose concentration curve types, and writing of statistical method sections and result sections related to curve modeling. Ian Thornell, Ph.D., ion transport specialist was located after primary data was gathered and provided invaluable information from the perspective of order of reactions, kinetics of GLUT transporters, the effects of osmolar differences, and synovial membrane diffusion characteristics that could explain the effects we saw on dextrose concentrations.

Data interpretation, Academic Writing:  David Rabago, M.D. of Penn State University provided scientific oversight, data interpretation, editorial or original writing on all parts of the manuscript.

Comment 11: The one small redeeming factor is that this is a difficult study to perform from ethical consideration and obtaining a consenting subject.

Response 11: Thank you. This was also more fully addressed previously (See Item 1).

Reviewer 2 Report

Comments and Suggestions for Authors

Dear Author

Your manuscript title, "Rapid decrease in dextrose concentration after intra-articular knee injection: Implications for mechanism of action of dextrose prolotherapy and concentration based finding" is a study on patient-based research work about the action of dextrose prolotherapy. 

The abstract has been written well with focus objectives

The introduction also organised a good compilation of relevant data.

Material and methods: Does the lead (first) author also have a patient?

Result: The result also has very interesting and exciting outcomes.

Limitation: The authors also mention the limitations as well

References: all references are relevant.

Author Response

Comment 1: The abstract has been written well with focus objectives

Response 1: Thank you.

Comment 2: The introduction also organized a good compilation of relevant data.

Response 2: Thank you.

Comment 3: Material and methods: Does the lead (first) author also have a patient?

Response 3: No. The lead author was the study participant in this N-of-1 study, which followed ethical and administrative standards of human subjects research. See Item 1, Reviewer 1.

Comment 4: Result: The result also has very interesting and exciting outcomes.

Response 4: Thank you.

Comment 5: Limitation: The authors also mention the limitations as well

Response 5: Thank you.

Comment 6: References: All references are relevant.

Response 6: Thank you.

Round 2

Reviewer 1 Report

Comments and Suggestions for Authors

The revised manuscript has taken out some of the unnecessary and redundant figures.  The manuscript probably also could do without fig 2 which presents the same data as in Table 1 and summary of combined data is shown in an additional figure.

Author Response

Comment 1:  The revised manuscript has taken out some of the unnecessary and redundant figures.  The manuscript probably also could do without fig 2 which presents the same data as in Table 1 and summary of combined data is shown in an additional figure.

Response 1:     We respectfully suggest retaining both figures because Figures 2 and 3 serve different purposes and reveal different aspects of the data. Figure 2 confirms how closely each dextrose concentration curve approximated the mathematical best fit pattern, which was either an s curve or an exponential pattern.  Figure 3 combined all data to indicate how data fit to a formulaic rate law; data indicated a 3rd order pattern of decline. Thus, figure 2 confirmed that dextrose concentration decline was rapid within each trial, and figure 3 indicated that more than two factors are required to explain the pattern of decline across all trials. Both figures have merit in our view. Please let us know if this clarification meets with your approval. If not, we are willing to discuss further. Thanks for this stimulating consideration of the data.